# Money attitudes, financial capabilities, and impulsiveness as predictors of wealth accumulation

Mark Fenton-O'Creevy[1]©*, Adrian Furnham[2]©

1 Open University Business School, The Open University, Milton Keynes, United Kingdom, 2 BI Norwegian School of Management, Oslo, Norway

© These authors contributed equally to this work.
* mark.fentonocreevy@open.ac.uk

**Data Availability Statement:** The full dataset is deposited with the UK Data Service (formerly the UK Data Archive), at https://doi.org/10.5255/UKDA-SN-8132-1. The owner of the intellectual

## Abstract

In this study we examined three correlates of personal wealth–financial capability, buying impulsiveness, and attitudes to money in a large UK adult sample (N = 90,184). We were interested in how these psychological variables related to personal wealth controlling for well-established demographic correlates: age, education, gender, and household income. We drew on three personal wealth variables based on savings and investments, property wealth and personal items. Using correlational and regression analysis we tested three specific hypotheses which each received support. Our variables accounted for around half the variance with respect to property value, and two thirds with respect to investments. The hierarchical regression onto the savings and investment factor showed two thirds of the variance was accounted for: the demographic variables accounted for 27% of the variance, money attitudes an additional 14%; financial capability an additional 24% and buying impulsivity no additional variance. Age, income, and planning ahead were the most powerful and consistent predictors of wealth variables, with associating money with security as an important predictor for savings and investments. Implications for helping improve financial literacy and capability are noted. Limitations are acknowledged.

## Introduction

Why are some people wealthier and "better off" than others? What of very rich people: how did they achieve their wealth [1]? Given similar income levels, what predicts who will build greater wealth over their lifetime? Why are people of the same age and similar background factors so different in their wealth in middle age?

These are simple questions with very complicated answers and the relevant academic disciplines have paid attention to quite different variables in attempting to give explanations. Sociologists and economists tend to focus on education and social class, or on presumed differences in discount factors applied to the utility of delayed consumption. Although some have paid attention to more attitudinal factors, such as the propensity to plan and to financial knowledge [e.g., 2], the majority explore socio-political and economic factors to explain personal wealth accumulation.

property in the dataset, the British Broadcasting Corporation (BBC), has imposed some conditions on data access which were also imposed on the original researchers. The Data Collection is available to users registered with the UK Data Service, [registration is free]. Access is limited to applicants based in HE/FE institutions, for not-for-profit education and research purposes only. Users of the data set are subject to identity verification by the UK Data Service and are required to agree to the conditions of use before downloading.

**Funding:** MFOC & AF: The British Broadcasting Corporation (BBC) provided substantial support 'in kind' organising promotion of the survey, sample recruitment, the data collection platform and data validation at no cost to the researchers. They offered general advice on the survey design (out of their previous experience of supporting this kind of survey) but final study design rested with the authors and the BBC had no role in the analysis, decision to publish nor preparation and writing of this manuscript. The primary goal of the BBC in supporting and promoting this research lay in supporting public engagement with social science in line with their public service mission.

**Competing interests:** The British Broadcasting Corporation (BBC) provided substantial support 'in kind' organising promotion of the survey, sample recruitment, the data collection platform and data validation at no cost to the researchers. They offered general advice on the survey design (out of their previous experience of supporting this kind of survey). There are no patents, products in development or marketed products associated with this research to declare. This does not alter our adherence to PLOS ONE policies on sharing data and materials.

Psychologists on the other hand have concentrated on individual differences. They have been concerned to explain how people from similar socio-economic backgrounds and in the same region and country have ended up with very different incomes and wealth accumulation. In this study we draw on secondary analysis of an existing large dataset to examine four classes of possible predictors of wealth: demography (age, education, gender, income); financial capability; money attitudes, and buying impulsiveness. Whilst it is recognized that demographic factors are usually the most powerful determinants of wealth, we are interested in the incremental explanatory validity of our three chosen psychological factors.

## Financial capability and knowledge

There is vast, fast-growing, and inter-disciplinary literature on financial knowledge and competence. The determinants of adult financial capability, distress, knowledge and financial well-being, are important as they can have a considerable impact on a person's health and welfare [3]. There has been a great deal of interest around the world on the important topics of financial wellbeing, capability [4], risk-taking [5]and satisfaction [6,7]. Many have looked at attitudinal, demographic, and financial literacy correlates of financial factors, some concentrating particularly on young people [8–10].

Various authors have concentrated on how to measure financial knowledge [11] and there remains no consensus as to a clear operational definition of (adequate, appropriate) financial knowledge. There is also significant variation between authors in definitions of financial literacy [12] and financial capability, which are often taken to be synonyms or to significantly overlap [13]. For example the OECD defines financial literacy as "*A combination of financial awareness, knowledge, skills, attitudes and behaviours necessary to make sound financial decisions and ultimately achieve individual financial well-being*" [14], whilst the Financial Capability Strategy for the UK [15: p6], defines financial capability as "*people's ability to manage money well, both day to day and through significant life events, and their ability to handle periods of financial difficulty*". Drawing on a systematic review Goyal and Kumar (13: p81) suggest that "[p]*eople can be financially literate when they have knowledge, understanding and skills to take care of their personal finances but they cannot be called financially capable unless it is reflected in their actual behaviour*"; and "*financial capability is the manifestation of a specified level of financial literacy and the execution of a desired financial behaviour*" (p95).

There has been a great cross-disciplinary (economics, finance, psychology) interest in both financial literacy and capability. Indeed, so much so that there have been a number of recent reviews and meta-analyses [16]. In an important recent systematic review Goyal and Kumar [13] identified three major themes: levels of financial literacy amongst distinct cohorts, the influence that financial literacy exerts on financial planning and behaviour, and the impact of financial education. They also identified other themes including financial capability (overlapping with financial literacy but typically also considering attitudes and behaviours), financial inclusion, gender gap, tax and insurance literacy, and digital financial education.

Von Stumm et al. [17] found that socioeconomic status was associated with financial capabilities but not with money attitudes, which were independent of financial capabilities. Whilst they found poor financial capabilities to be greater risk factors for adverse financial outcomes than money attitudes; they concluded the latter important and likely targets to be changed by education and other interventions.

In this study we examine the relationship of five facets of financial capability, identified in prior research [18], to personal wealth. As we go on to discuss, we also consider the relationship between money attitudes, buying impulsiveness and wealth.

## Money attitudes

There has been a great deal of work in the last decade concerning attitudes to, and beliefs about, money [19]. Researchers have identified a range of money attitudes that underpin differences in individual orientations to money and its uses [7,20–22]. Money attitudes have been linked with many *demographic* variables [23]. Studies have found money attitudes related to *gender* [7,21,22,24], *culture* [25–27], *education level* [28,29], and *political and religious value*s [7,22].

The analysis we report in this paper examined four attitudes associated with money, following an approach developed by Furnham et al. [22]. Many popular writers have attempted to differentiate money types such as the hoarder, spender, binger, monk, avoider, amasser, worrier, risk taker, and risk avoider [e.g., 30–32]. The four most common unique money-associated goal orientations have been identified: Security, Power, Love and Freedom [21] and it is these we focus on in the current study.

Having and saving money, for many, can stand for *Security*. Money is a kind of emotional lifejacket, a security blanket, a method to stave off anxiety. These individuals tend to be savers, collectors, and self-deniers and can be distrustful of others [32]. Having money reduces dependence on others and vulnerability to adverse events, thus reducing anxiety.

For some, money also represents *Power* and prestige because money can buy goods, services, and loyalty. It can be used to acquire importance, domination, and control. Psychoanalysts argue that money and the power it brings can be understood as a search to regress to infantile fantasies of omnipotence [33]. For others, money can represent *Freedom* or autonomy, allowing them to escape the boredom of routine or unwelcome obligations and constraints. Finally, money may also stand for *Love* or generosity, enabling the buying and selling of intimacy and affection or as a way of caring for those close to them [17].

Over the past decade there has been an increasing interest in assessing money attitudes and examining their causes and consequences, particularly for individuals' financial well-being.

## Buying impulsiveness

There is a well-established clinical literature on the trait impulsivity, and particularly impulse buying and spending [34]. It is seen as a cause of considerable economic distress and difficulty [35]. Wealth generation may inevitably be impeded by excessive spending. A key element of such behaviour is 'hedonic purchasing' to satisfy immediate impulses. That is, impulsive buying, defined by Rook and Fisher [36] as a consumer's tendency to buy spontaneously, unreflectively, and immediately.

There is evidence that impulsive buying tendencies are associated with greater propensity to experience financial distress including a greater probability of bankruptcy [35]. Other adverse outcomes of impulsive buying have been shown to include debt, depression, and marital discord [37].

Thus, in this study we examined the association of this trait on wealth accumulation

## This study

While income over a life course clearly represents opportunities to accumulate wealth, and age represents the time available in which to accumulate wealth, other factors may determine whether these opportunities are realized. We know that, traditionally, men have been expected to be primary income generators in the family, women having typically less access to the economic and social elements of life that foster wealth accumulation, and that therefore men often have greater personal wealth than women [38]. We also know that education is likely to be an important correlate of wealth [2] as it has important relationships to occupational status

and success. Thus, we expect income, age, gender and education to be important predictors of personal wealth. We will consider these as control variables, as our interest is in the incremental validity of the three sets of variables discussed above.

In particular, this study is concerned with factors that will either influence propensity to save, invest and preserve wealth, or that will influence good judgement in financial matters and propensity to avoid significant financial losses.

Our first hypothesis concerns financial capabilities. Given the evidence, reviewed above, that financial capability predicts sound financial judgement it seems likely that accumulation of greater wealth will be predicted by higher levels of financial capability. Thus, we hypothesize: -

*H1: Higher levels of financial capability will be associated with higher levels of wealth.*

Second, studies of money attitudes have found significant relationships between money attitudes and a variety of financial outcomes, including difficulty meeting financial commitments, and adverse financial events ranging from denial of credit to bankruptcy [e.g., 39].

There is a consistent theme in these studies that those who associate money with security, but not generosity/love, freedom and power/status seem to exhibit more careful financial management and monitoring of spending [19]. These are behaviours likely to ensure money can be retained from income to accumulate different forms of wealth. Thus, we hypothesize:

*H2a: Money attitudes will explain significant variance in accumulated wealth.*

Specifically of the four dimensions of money attitudes considered above we predict:

*H2b: A stronger orientation to seeing money as security will be associated with greater accumulated wealth.*

Third, we have argued that wealth generation may be impeded by excessive spending associated with buying impulsiveness. Thus, we hypothesize: -

*H3: Higher buying impulsiveness will be associated with less accumulated wealth.*

## Methods

We test the hypotheses outlined above though secondary analysis of an existing data set [40], archived by the UK Data Service (formerly UK Data Archive) and available to researchers via free registration with the UK Data Service. In the original study, conducted in the UK in collaboration with the British Broadcasting Corporation (BBC), participants took part in an online BBC survey (promoted on a flagship prime-time consumer affairs television program and on radio programs) on their relationships with their money and their financial behavior. This was part of a series of online scientific investigations and data were collected across 6 weeks from survey launch.

### Ethics and data availability

The original study received research ethics approval from the Open University Human Participants and Materials Research Ethics Committee (approval reference: HPMEC/2010/#794/1).

Participants' informed consent in the original study was managed by requiring them to read online information on the purposes of the survey, the ways in which their personal data would be protected and the research use and storage of their anonymized data on a public research data repository. Participants were told that by participating they would help scientists understand how and why different people think and feel about money in different ways. As an incentive for participation, they were offered (automated) video and web feedback on key self-reported financial capability measures, and their score on a financial knowledge test on completion of the online questionnaire, followed by a video of a television presenter offering them tips on personal financial management.

The first screen of the survey gave details of the study, anonymization of the data, the procedure for withdrawing consent and the use of the data in research. The survey required indication of informed consent before proceeding. Participants informed consent was indicated by ticking a box to agree to participate subject to the information they had read on the use of their data. Participants failing to agree were exited from the survey. Initial screening by the BBC excluded participants under the age of 16 from registering for surveys in this series. This is the age from which UK law assumes capacity to consent on a range of matters including to medical treatment and research such as clinical trials without parental involvement. All participants were provided with the email address of the BBC data controller and advised that they could withdraw their consent and have their data deleted by contacting the data controller.

The full dataset is deposited with the UK Data Service, at https://doi.org/10.5255/UKDA-SN-8132-1. The owner of the intellectual property in the dataset, the BBC, has imposed some conditions on data access which were also imposed on the original researchers. These are:

"*The Data Collection is available to users registered with the UK Data Service,* [registration is free]. *Access is limited to applicants based in HE/FE institutions, for not-for-profit education and research purposes only. The BBC Data Collection shall not be used in a manner which*:

- *distorts the original meaning of the BBC Data Collection, for example by changing the context;*

- *discriminates against any specific social group or otherwise exploits vulnerable sections of society;*

- *promotes, encourages or facilitates violence;*

- *promotes, encourages or facilitates illegal activity;*

- *promotes, encourages or facilitates terrorism or other activities which risk national security;*

- *promotes the tobacco, armaments, alcohol or pornographic industries;*

- *encourages hatred on grounds of race, religion, gender, disability, age or sexual orientation;*

- *promotes, encourages or facilitates anti-social behaviour;*

- *might be perceived as damaging the BBC's reputation for accuracy or impartiality; or*

- *otherwise brings the BBC into disrepute.*"

Users of the data set are subject to identity verification and are required to agree to the conditions of use before downloading.

The data set contains no identifying information on participants such as name, address, IP address, specific dates or contact information. The full data set does contain partial postcode data (last two characters are truncated), sufficient to identify postcode sector, which would typically cover around 3000 addresses.

## Participants

The data set includes data on 109,472 participants, 16 or older. The present study focused on participants with full data on wealth and income. After deletion of cases without this data, there remained 90,184 participants of which 52.2% identified as female. 10.7% % were between 16 and 24 years old, 29.5% between 25 and 34 years old, 25.5% between 35 and 44 years old, 18.1% between 45 and 54 years old, and the remainder, 14.3%, 55 and older. 91.6% classified themselves as white British. 43.3% had secondary school qualifications or lower, 40.8% had an undergraduate degree or equivalent and 15.9% a postgraduate degree. 23.1% had household

income of less than £20,000 p.a., 34.6% between £20,000 and £39,999 p.a., 30.4% between £40,000 and £74,999 p.a. and 11.9% earned £75,000 or more. Geographical data (partial post-code) suggested a reasonably even distribution of responses across the UK relative to population densities. However, given the online nature of the survey there may have been under sampling amongst those with very low income and the elderly.

## Dependent variables (DVs)

We draw on three measures of wealth from the dataset which were assessed through the following questions: -

*Property*: "If you own your own home, what do you think its value is less any mortgage you have? If you have more than one property, include the value of all your properties, less any mortgage." (1 = "£0 or less", to 6 = £500,000 or more).

*Savings and investments*: "If you have any savings and other financial investments (such as bank and building society accounts, unit trusts, insurance bonds with a cash in value, shares and so on), what do you think is the value of these savings and investments? Please note this value must be minus any money you owe on credit cards, personal loans, or other debts, but you do not need to deduct your mortgage. Do not include the value of any pensions or money in any pension schemes." (1 = less than £0, to 8 = £50,000 or more).

*Physical items*: "If you have any physical items that you think of as part of your wealth (e.g. car, caravan, artwork, jewelry, gold, valuable antiques, wine held as an investment, and so on) what do you think is the value of these items?" (1 = £0 or less to 8 = £50,000 or more).

The dataset does not include a measure of pension wealth, in part due to the complexity for participants in accurately estimating such assets.

## Control measures

We have argued that greater income provides greater opportunity to accumulate wealth and greater age represents greater time in which to accumulate wealth. Both have been shown to have significant and substantial associations with wealth measures in prior research [e.g.,41]. There is also evidence of significant gender effects on financial outcomes including both income and wealth [e.g., 38]. Finally, education has been linked to financial outcomes. For example, Cole et al. [42] find higher levels of education level (but not personal finance education) to be associated with greater investment income and equities ownership. Thus, in our analysis we control for Age (in years), Household Income in the last year (in effect, we treat current income as a proxy for lifetime earnings), eight bands (1 = up to £9,999 per annum, to 8 = £150,000 or more per annum), gender (0 = male, 1 = female), and education level (1 = did not complete GCSE / CSE / O-Levels or equivalent, to 6 = postgraduate degree).

## Financial capabilities

Five financial capabilities were assessed by sets of three to eleven questions each spanning multiple-choice and Likert-type items derived from the highest loading items in factors identified in a large-scale study by Atkinson and colleagues [18]. Items include both behavioural and attitudinal questions. For full scale details see [17].

Capability scales, with sample items were:

**Making ends meet.** "In the last 12 months, how often have you run out of money before the end of the week/month or needed to use your credit card or overdraft to get by?" (reversed), 4 items, Cronbach alpha ($\alpha$) = .77). Higher scores indicate better ability make ends meet.

**Keeping track.** "How accurately do you know how much money you have either in your current account or, if you don't use a current account day to day, how much cash you have in hand?" 4 items, α = .82.

**Choosing products.** "To what extent do you (or you and your partner) normally shop around when you open or take out a financial 'product'? (Products such as a bank account, credit/store card, insurance, loan, insurance.)" 3 items, α = .61.

**Planning ahead.** "For how long do you think you could still make ends meet if you lost your main source of income?" 4 items, α = .93.

**Staying informed.** "How important is it for people like you to keep up with changes in prices and to look out for deals on goods and services?" 11 items, α = .59.

## Money attitudes

This 16-item set of four scales [21] assesses attitudes to money, with questions rated on a Likert scale ranging from 1 (strongly disagree) to 5 (strongly agree). The four scales are: "money as security" (α = .63); "money as freedom" (α = .61); "money as power (α = .73); and "money as love "(α = .64).

Sample questions for each scale are:

## Security

"I'd rather save money than spend it."

## Freedom

"With enough money, you can do whatever you want."

## Power

"Money is important because it shows how successful and powerful you are."

## Love

"I often demonstrate my love to people by buying them things."

## Buying impulsiveness

Rook and Fisher's [36] original nine-item scale was shortened due to space constraints in the survey, choosing the five items loading most strongly on the single factor in the original study. Items "I often buy things without thinking," "I often buy things spontaneously," "'I see it, I buy it' describes me", "'Just do it' describes the way I buy things", "'Buy now, think about it later' describes me" (α = 0.90).

## Analysis

We first calculated Pearson correlations between all variables. Second, to account for relationships between the DVs we conducted multivariate multiple regression with the control variables, financial capability variables, money attitudes, and buying impulsivity entered as independent variables (IVs) and the three wealth variables entered jointly as DVs.

Multivariate multiple regression is an analysis method for modeling multiple DVs with a single set of predictor variables. It is especially useful where the DVs have significant intercorrelations; as, in this model [43]. This method produces significance tests for the predictors of the DVs which control for all other relationships in the model, including via the other DVs

[44]. This analysis was carried out using the multivariate version of the general linear model procedure in SPSS 25.

## Results

Table 1 shows Pearson correlations between all pairs of variables (with means and standard deviations on the diagonal).

Since the data set is very large even quite small correlations are significantly different from zero. The correlations between all independent variables and the three dependent (wealth) variables are significant at the p < .05 level with most significant at p < .001. However, there are interestingly different patterns of association with different forms of wealth.

Turning first to the control variables, we find that as expected greater age is associated with greater wealth of all three types measured: showing the strongest correlation (.58) with property wealth and the lowest (.23) with possession of valuable physical items.

Education shows rather modest, although significant positive correlations with types of wealth, the strongest being with savings and investments (.13). Gender (being female) shows small negative correlations with wealth; weaker for property wealth (-.06) and a little stronger for savings/investments (-.16) and physical items (-.15).

**Table 1. Pearson correlations means and standard deviations of all variables.**

| | 1 | 2 | 3 | 4 | 5 | 6 | 7 | 8 | 9 | 10 | 11 | 12 | 13 | 14 | 15 | 16 | 17 |
|---|---|---|---|---|---|---|---|---|---|---|---|---|---|---|---|---|---|
| 1.Property | 2.91 (1.59) | | | | | | | | | | | | | | | | |
| 2.Savings /investments | .52*** | 4.63 (2.49) | | | | | | | | | | | | | | | |
| 3.Physical items | .40*** | .39*** | 3.98 (1.80) | | | | | | | | | | | | | | |
| 4.Age | .58*** | .41*** | .23*** | 39.86 (13.33) | | | | | | | | | | | | | |
| 5.Education | .03*** | .13*** | .07*** | -.15*** | 4.21 (1.46) | | | | | | | | | | | | |
| 6.Female | -.06*** | -.16*** | -.15*** | -.04*** | .00 | 0.52 (0.50) | | | | | | | | | | | |
| 7.Income | .33*** | .29*** | .32*** | .03*** | .30*** | -.14*** | 4.13 (1.86) | | | | | | | | | | |
| 8. Power | -.06*** | -.04*** | .03*** | -.11*** | -.06*** | -.16*** | .01*** | 2.14 (0.76) | | | | | | | | | |
| 9. Freedom | .02*** | -.01*** | .01* | .05*** | -.11*** | -.08*** | -.02*** | .44*** | 3.41 (0.86) | | | | | | | | |
| 10. Security | .12*** | .41*** | .11*** | .01*** | .09*** | -.08*** | .07*** | .06*** | .04*** | 3.21 (0.77) | | | | | | | |
| 11. Love | -.08*** | -.09*** | .02* | -.08*** | .01 | -.07*** | .02*** | .25*** | .13*** | -.06*** | 3.21 (0.94) | | | | | | |
| 12.Making Ends Meet | .36*** | .62*** | .29*** | .24*** | .12*** | -.13*** | .23*** | -.09*** | -.06*** | .42*** | -.13*** | 16.89 (3.44) | | | | | |
| 13.Keeping Track | -.03*** | -.06*** | -.02*** | .06*** | -.13*** | .04*** | -.13*** | -.01 | .01*** | .08*** | -.01*** | .07*** | 14.78 (3.05) | | | | |
| 14.Planning Ahead | .49*** | .77*** | .38*** | .34*** | .16*** | -.12*** | .33*** | -.08*** | -.05*** | .44*** | -.09*** | .72*** | -.01*** | 15.38 (4.02) | | | |
| 15.Choosing Products | .24*** | .28*** | .20*** | .18*** | .14*** | -.04*** | .17*** | -.10*** | -.07*** | .19*** | -.06*** | .35*** | .16*** | .37*** | 16.97 (2.91) | | |
| 16.Staying Informed | .27*** | .29*** | .22*** | .20*** | .11*** | -.14*** | .20*** | .03*** | 0.00 | .23*** | -.02*** | .30*** | .23*** | .35*** | .44*** | 15.73 (3.54) | |
| 17.Buying Impulsiveness | -.13*** | -.22*** | -.05*** | -.12*** | -.09*** | .14*** | -.03*** | .22*** | .15*** | -.31*** | .28*** | -.32*** | -.11*** | -.25*** | -.26*** | -.22*** | 6.43 (5.07) |

*** Correlation is significant at the 0.001 level (2-tailed).

** Correlation is significant at the 0.01 level (2-tailed).

* Correlation is significant at the 0.05 level (2-tailed).

Mean (std. deviation) on the diagonal. N = 90184.

(Household) income, as expected shows positive correlations with wealth measures, varying from .29 to .33.

Considering financial capabilities, the results offer initial support for H1 with planning ahead showing the strongest associations with wealth measures; .77 for savings/investments, .49 for property wealth and .38 for physical items. Making ends meet shows a particularly strong correlation with savings/investments (.62) and lower correlations for property wealth (.36) and physical items (.29). Keeping track shows the lowest correlations with wealth measures, interestingly, all negative, ranging from -.02 to -.06. This may suggest a causal direction from wealth to keeping track, the wealthier have less need to closely track their finances.

Regarding money attitudes, there is initial support for H2, with all attitudes showing significant correlations with wealth measures. As hypothesized (H2a) seeing money as security is significantly positively associated with all wealth measures, most strongly for savings/investments (.41) and less strongly for property (.12) and physical items (.11).

Providing initial support for H3, buying impulsiveness shows significant (although modest) inverse correlations with all three measures of wealth, correlating most strongly with savings/investments (-.22).

Table 2 provides an omnibus test of the overall multivariate multiple regression model, showing the significance of joint variance between each IV and the group of three (wealth) DVs.

The partial eta squared values are effect sizes that provide a comparative indication of unique common variance between each IV and the DVs. Partial eta squared ($\eta^2_p$) values may be understood as the proportion of unique variance in the DVs explained by each IV once all other modelled relationships are controlled for [45].

We use partial eta squared measures in this study as an indicator of the relative importance of each IV in explaining the DVs. The Hotelling's trace value (also known as Hotelling-Lawler trace) is used to test the null hypothesis for each IV that it explains zero variance in the DVs.

This statistic can be shown to generalise the partial F test used for ordinary least squares regression, with one dependent variable, to a test for regression with multiple dependent variables [46].

**Table 2. Multivariate tests of multivariate multiple regression with three wealth measures.**

| Effect | Hotelling's Trace | F | Sig. | Partial Eta Squared |
|---|---|---|---|---|
| Intercept | .080 | 2417.43[a] | .000 | 0.074 |
| Age | .407 | 12230.81[a] | .000 | 0.289 |
| Education | .005 | 139.22[a] | .000 | 0.005 |
| Female | .018 | 538.85[a] | .000 | 0.018 |
| Income | .109 | 3271.23[a] | .000 | 0.098 |
| Power | .003 | 77.73[a] | .000 | 0.003 |
| Freedom | .001 | 20.07[a] | .000 | 0.001 |
| Security | .034 | 1028.78[a] | .000 | 0.033 |
| Love | .004 | 106.57[a] | .000 | 0.004 |
| Making Ends Meet | .016 | 467.27[a] | .000 | 0.015 |
| Keeping Track | .013 | 391.69[a] | .000 | 0.013 |
| Planning Ahead | .354 | 10631.37[a] | .000 | 0.261 |
| Choosing Products | .004 | 108.56[a] | .000 | 0.004 |
| Staying Informed | .006 | 177.88[a] | .000 | 0.006 |
| Buying Impulsiveness | .003 | 76.18[a] | .000 | 0.003 |

a. Exact statistic. Hypothesis d.f. = 3, error d.f. = 90,167.

The Hotelling's trace value is significant at p < .001, for all variables suggesting all IVs explain non-zero variance in the wealth measures. Consistent with the correlation results, partial eta squared values show the most unique variance to be explained by age (.289), planning ahead (.261) and household income (.098).

Table 3 shows the regression parameters for each regression using the three different wealth measures as dependent variables. With other variables controlled for, the different pattern of relationships between independent variables and wealth measures becomes more apparent.

First, looking at the control variables we find age predicts all three measures of wealth, with the strongest association again being between age and property wealth, but predicting more unique variance in property wealth than in the other two measures. Education shows a positive association with investment/savings but now shows an inverse association with property wealth and physical items. Gender (being female) shows an inverse association with savings/investments and physical items but a positive association with property wealth. Household income shows a positive association with all three wealth measures but predicts substantially less unique variance in investments/savings than the other two measures.

The results offer partial support for H1, in that the financial capabilities of planning ahead and making ends meet show positive associations with all three measures of wealth (in both cases they explain most unique variance in savings/investments. However, the picture for the other financial capabilities is more complex. Keeping track shows a modest inverse relationship with property wealth and savings/investments and a very small but significant positive association with physical items. Choosing products shows a small but significant positive association with property wealth and physical items but a modest though significant inverse association with savings/investments. Staying informed shows a modest positive association with property wealth but no significant association with savings/investments.

**Table 3. Multivariate multiple regression parameters for each wealth measure.**

| Parameter | Dependent Variable | | | | | |
|---|---|---|---|---|---|---|
| | Net value of home and other property | | Savings and other financial investments (non-pension) | | Value of physical items | |
| | Unstandardised regression parameter (B) | Partial eta squared | B | Partial eta squared | B | Partial eta squared |
| Intercept | -1.88*** | .015 | -4.17*** | .062 | -.28*** | .000 |
| Age | .06*** | .225 | .04*** | .089 | .02*** | .016 |
| Education | -.02*** | .001 | .06*** | .003 | -.02*** | .000 |
| Female | .13*** | .003 | -.26*** | .007 | -.27*** | .006 |
| Income | .20*** | .064 | .06*** | .005 | .20*** | .042 |
| Power | .04*** | .001 | .05*** | .001 | .11*** | .002 |
| Freedom | .03*** | .000 | .00 | .000 | -.03*** | .000 |
| Security | -.04*** | .000 | .39*** | .028 | -.09*** | .001 |
| Love | -.05*** | .001 | -.03*** | .000 | .07*** | .001 |
| Making Ends Meet | .01*** | .001 | .08*** | .015 | .02*** | .001 |
| Keeping Track | -.02*** | .003 | -.05*** | .011 | .00* | .000 |
| Planning Ahead | .09*** | .029 | .35*** | .251 | .10*** | .026 |
| Choosing Products | .01*** | .000 | -.02*** | .001 | .03*** | .002 |
| Staying Informed | .02*** | .003 | -00 | .000 | .03*** | .002 |
| Buying Impulsivness | .00 | .000 | .01*** | .001 | .02*** | .002 |
| $R^2$ (sig)[a] | .49 (.000) | | .66 (.000) | | .23 (.000) | |

a. Adjusted $R^2$ not shown as not different to unadjusted figure given large sample size.

The results offer support for H2. Money attitudes explain significant unique variance in the three measures of wealth. However, H2a is only partially supported. Seeing money as security shows a positive association with savings/investments (explaining about 3% of unique variance). However, it shows a modest inverse association with property wealth and physical items. Money as power shows a modest positive association with all three measures of wealth. Money as freedom shows a modest positive association with property wealth but a modest inverse association with physical items. Money as love shows a modest inverse association with savings investments, but a modest positive association with physical items.

Turning to H3, despite inverse correlations with all three wealth measures, buying impulsivity explains no unique variance in property wealth and has a modest positive association with savings/investment and physical items. Thus, H3 is unsupported.

Given the discrepancy between correlational results and regression results and noting the correlations of buying impulsivity with financial capabilities and attitudes, we conducted a supplementary multivariate multiple regression analysis. We entered buying impulsiveness as a predictor variable with just the control variables. The results showed significant negative regression parameters for buying impulsiveness for property wealth (-.02**) and savings/investments (-.07***) but no significant parameter for physical items. This suggests buying impulsiveness may be associated with reduced investment in property and other savings, but with the effect masked by the variance it shares with financial capability and attitudes.

Finally, a series of hierarchical regressions were computed to look at the relative contribution of the four sets of predictor variables: demographic controls, money attitudes. financial capabilities and finally buying impulsivity. For the first wealth measure (property wealth) the demographics accounted for 44% of the variance, money attitudes an addition 1%; financial capability an additional 5% and buying impulsivity no additional variance. For the second wealth measure (savings/investments) the demographics accounted for 27% of the variance, money attitudes an addition 14%; financial capability an additional 24% and buying impulsivity no additional variance. For the third wealth measure (physical items)) the demographics accounted for 16% of the variance, money attitudes an addition 1%; financial capability an additional 5% and buying impulsivity no additional variance.

## Discussion

Having access to this unique data set allowed us to explore some hitherto neglected variables which we believe contribute to the accumulated wealth of adults. We had three dependent variables all associated with personal wealth which correlated .39 < r < .52. Although the findings are somewhat incremental in relation to prior studies, the size of the sample adds considerable weight to previous findings. The ability to differentiate effects of financial capability, buying impulsiveness and money attitudes on different forms of wealth is also an important contribution to debates about the correlates of wealth accumulation.

The results of major interest were in the regressions shown in Table 3. Age, education, male gender and household income all had positive associations with wealth. As may be expected our four demographic control variables accounted for much of the variance particularly age. However, what is of particular interest are the psychological attitude, belief and behavioral variables that we investigated. The hierarchical regressions showed that for the three dependent wealth variables the demographic (control) factors accounted for between 16% and 44% of the variance and money attitudes added between 1 and 14%, financial capabilities between 5 and 24% but impulsive buying explained no unique variance.

We were particularly interested in three predictor variables. First, we had different measures of financial capability the most important of which was planning ahead. This could be

seen as a fundamental aspect of all money management, allowing for better money decisions. Planning ahead may also be related to postponement of gratification, the opposite of impulsiveness. Whilst it is possible that some people become obsessed with monitoring their own monetary situation and that of the market, planning ahead is clearly fundamental to all money management [32].

The four money attitude variables showed three things. First, each was related to the different measures of wealth in small but important and understandable ways. Thus, those who associated money with love had more valuable objects, but less valuable property wealth. Those who associated money with security had greater investment wealth but less physical and property wealth, suggesting an emphasis on more liquid assets. Associating money with freedom seemed least associated to our wealth measures. Second, the attitudes were differentially related to the three wealth variables, even money as security which in previous studies shown to be the most healthy and adaptive association. Third, the money attitude questionnaire is a short but sensitive indicator of a range of financial variables which also gives insight for those interested in changing money beliefs and behaviors [47].

We did not find support in the regression analysis for our third hypothesis, about buying impulsiveness, though all the correlations were negative as hypothesised and further analysis suggested effects to be masked by covariance with other independent variables. Impulsivity in all spheres is associated with immaturity, poor emotional regulation and negative social outcomes [48]. It may be that effects of buying impulsivity on wealth are indirect, for example, accounted for via planning ahead, making ends meet, and choosing products with which it (inversely) correlates.

The patterns of difference are interesting between the three wealth variables, for example, indicating that money as security supports saving behavior but may reduce investment in property (perhaps by reducing propensity to risk overstretch on a mortgage) and may reduce acquisition of expensive physical items. Of the three dependent wealth factors our psychological variables accounted for most of the variance in the measure of *savings and investments* which is most related to systematic money management. This indicated that the psychological variables, which may be seen as "money-related habits" played a crucial role.

The results are interesting for those involved in financial sales, advice, planning counselling and education. Perhaps too much interest has been paid to the single variable of client risk-appetite within personal finance and not to other related attitudes and behaviors. Certainly, these results give insight into financial capability education, which is seen to be important in all aspects of money management and as much about avoiding poor money management as accumulating great wealth. The correlational data suggested that making ends meet (living within one's budget), planning ahead (for investments and expenditure) and staying informed are very important financial activities. For financial advisors, counsellors, planners and sales roles these should be relatively easy to assess, and for educators straightforward to teach. Providing people with simple-to-use Apps seems a useful start to help people of all ages achieve better money management. The money attitude results also suggest the importance of education going beyond building knowledge and key skills to engage learners in thinking about their relationships with money and spending and how to build attitudes conducive to financial wellbeing.

This, like all studies, had various limitations we are conscious of. First, some financial capability and money attitude measures had modest Cronbach alpha reliability scores. Internal reliability tends to be lower for shorter scales since they tend to have greater random error of measurement. Larger more heterogeneous sample sizes also generate lower reliabilities (perhaps reflecting greater ease of tailoring item language to smaller more homogeneous samples) [49].

The scales were measured as part of a large omnibus survey which had to trade off scale length for response rate and number of constructs measured. Clearly greater random measurement error (noise) can reduce statistical power of tests. However, given the large sample size, this is perhaps of less concern in the present study. Although the strength of relationships (effect size) may be attenuated.

Second, despite our large sample it was somewhat skewed to better educated and well-paid individuals, and restricted to participants living in the UK, though we believe the results would replicate on a more representative sample. Further research could usefully seek to replicate findings in different national settings.

Third, the cross-sectional nature of the data, does not allow attributions of causality. For example, it is not possible to rule out reciprocal relationships between ability to make ends meet and wealth. On the one hand, making ends meet should increase availability of funds to invest, conversely though, having savings as a buffer may reduce the need to rely on credit. Future studies might usefully collect longitudinal panel data to explore the causal patterns in these variables.

Finally, it would, as always, be useful to have more details about each individual's financial status (e.g. pension savings) so that our dependent variables were more comprehensively and sensitively measured, and our income measure gave some measure of earnings history. It would also be of interest to know how subjectively people thought they were wealthy, which gives some insight into their social comparison processes. It would also have been interesting to get some additional information such as the "high-yield" question concerning whether respondents saw themselves as spenders or savers [50]. For future users of this dataset, there is scope to extend the range of questions explored by exploiting the (partial postcode) geolocation data (at UK postal sector level) as this offers the possibility of combination with data, for example, on measures of relative social deprivation in different areas.

## Author Contributions

**Conceptualization:** Mark Fenton-O'Creevy, Adrian Furnham.

**Data curation:** Mark Fenton-O'Creevy.

**Formal analysis:** Mark Fenton-O'Creevy.

**Methodology:** Mark Fenton-O'Creevy.

**Visualization:** Adrian Furnham.

**Writing – original draft:** Mark Fenton-O'Creevy, Adrian Furnham.

**Writing – review & editing:** Mark Fenton-O'Creevy, Adrian Furnham.

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
