## [Decision Letter · Decision Letter 0]

10 Aug 2022

PONE-D-22-06524Money attitudes, financial capabilities, and impulsiveness as predictors of wealth accumulation.PLOS ONE

Dear Dr. Fenton-O'Creevy,

Thank you for submitting your manuscript to PLOS ONE. After careful consideration, we feel that it has merit but does not fully meet PLOS ONE’s publication criteria as it currently stands. Therefore, we invite you to submit a revised version of the manuscript that addresses the points raised during the review process.

We look forward to receiving your revised manuscript.

Kind regards,

Andrew T. Carswell

Academic Editor

PLOS ONE

https://journals.plos.org/plosone/s/file?id=ba62/PLOSOne_formatting_sample_title_authors_affiliations.pdf".

Reviewers' comments:

Reviewer's Responses to Questions

**Comments to the Author**

1. Is the manuscript technically sound, and do the data support the conclusions?

Reviewer #1: Yes

Reviewer #2: Partly

2. Has the statistical analysis been performed appropriately and rigorously? 

Reviewer #1: Yes

Reviewer #2: I Don't Know

3. Have the authors made all data underlying the findings in their manuscript fully available?

Reviewer #1: Yes

Reviewer #2: No

4. Is the manuscript presented in an intelligible fashion and written in standard English?

Reviewer #1: Yes

Reviewer #2: Yes

5. Review Comments to the Author

Reviewer #1: The paper is generally well written, and will make an important contribution to the field. There are however some areas where a few changes/adjustments can further strengthen the contribution of this manuscript. I have attached my reviewer report separately.

Reviewer #2: Overall, I felt that the authors have put together a pretty compelling manuscript that provides some good story lines. Here are some comments that might help push this manuscript along.

You might get some pushback on the notion that financial capability and financial literacy amount to the same thing, as you do on page 3 of your manuscript. Can a financially capable person not be very financially literate as experts measure the concept?...I would probably think so. Can a financially literate person be not very financially capable in a lot of settings, most likely due to low financial means?...I would think most definitely. You might want to check with some of the fairly nuanced literature on this subject matter through financial planning journals that are out there.

While I don’t have any quibbles with this author’s current lit review sources, there are a number of authors in the U.S. who specialize in the concept of financial literacy and personal finance that could probably serve as references here. Regarding the former, please check out some of the work from Brenda Cude and Gianni Nicolini (and many others). Some of the journals that you might want to scour for more perspective include the Journal of Consumer Affairs, Journal of Personal Finance, Journal of Financial Planning and Counseling, and Financial Planning Review. Two other book resources that should prove helpful for you are Degruyter’s Handbook of Personal Finance and the Routledge Handbook of Financial Literacy.

On page 5, you mention that “many popular writers have attempted to differentiate…” without providing a list of multiple references. Please fix this.

I would also make sure that all of your statements of fact are backed up by citations. I caught a few of them, but they are probably too numerous to enumerate in this space. For example, check out the top of page 7 with the statements “we know that…” These are practically begging for citations.

While I am OK with the Hypothesis 2a on page 8, the write-up leading up to the hypothesis throws me off just a little bit. It seems that you are throwing a lot of words and concepts in there that are to be associated with “financial attitudes” that I might question.

Pretty impressive data set, and one that I admit I am not familiar with. Obviously the amount of data used is large enough to produce some interesting results. I am a little confused in the language on page 8 regarding the research methods, specifically regarding the data. While you refer to it as a secondary data set, you say that it required “informed consent”. This would be unlike a lot of large secondary data sets that exist in the U.S. Can you elaborate further? BTW, kudos for mentioning that “under sampling amongst those with very low income and the elderly”. I would have guessed that, but you performed due diligence in pointing that out to your audience.

You mention “five bands” of income within the data set, but you list up to eight in your descriptive example. Please elaborate further or correct that item.

The scale you mention on pages 10 to 11 appears to be more than 16 items, although I could be wrong about that. Please check though, just to make sure. As an aside, I think that a table detailing the development of the scales might be a worthwhile venture here.

Regarding some of your findings, there are a few things that do not seem to add up intuitively. The high correlation between making ends meet and savings/investment does not seem to add up. If you are scrounging to make a living month to month, how would you have enough to invest and put away? I do like your insights on the keeping track front, namely that wealthy folks may not need to track given their situation.

I am not entirely familiar with the analysis you have conducted in Table 2. Can you possibly explain this methodology a little more? I have never worked with a model that I had to explain partial eta squared measures or Hotelling’s trace values.

Does the inverse association between property wealth and education mean that educated people know not to invest in the property market? That finding would raise a few eyebrows, I would imagine.

Be careful with using the following statement in your discussion, “Older, better educated males in higher income households had most wealth”. This would imply that you are using an interactive variable combining all three elements, when in truth you are only looking at them individually.

I am not so sure that it is surprising that gender played as big a role as it did in wealth building. This could be confirmed by canvassing a lot more of the financial planning literature as I suggested earlier. There should be some good gender-related research on this kind of subject matter. There are still some pretty entrenched male-dominated financial planning gender roles within households that have endured throughout decades. Family and consumer sciences literature can attest to that as well.

I would not beat yourself up over the lack of findings regarding the impulsiveness angle. It is an interesting facet of personal finance. Sometimes not finding statistically results is a story unto itself.

While you have referred to financial education, you have not really referred to financial planners or financial counselors, which are professions that have a toehold within the U.S. I would do some research on those professional lines, because this particular research has some implications for these professions.

One other line of research that you might want to address is that of financial risk tolerance, which seems to have some relevance with some of your scalar sub-items. John Grable is a researcher that has written a ton on this front. I am specifically thinking along the impulsivity front, and its element of financially risky behavior.

There are some typos throughout the document. These don’t distract from the author’s generally good writing style, but it might not hurt to have a copy editor go through the document all the same.

6. PLOS authors have the option to publish the peer review history of their article (what does this mean?). If published, this will include your full peer review and any attached files.

Reviewer #1: No

Reviewer #2: No

---

## [Author Response · Author response to Decision Letter 0]

11 Oct 2022

*We are grateful to the editor and reviewers for the constructive and helpful comments. We set out in detail how we have addressed these below.

JOURNAL REQUIREMENTS:

*We have now sought to ensure that we have met these. If we have erred in any respect, we will be happy to correct this.

*We now include much more detailed information on the informed consent process:

“The original study received research ethics approval from the Open University Human Participants and Materials Research Ethics Committee ( approval reference: HPMEC/2010/#794/1). 

Participants’ informed consent was managed by requiring them to read online information on the purposes of the survey, the ways in which their personal data would be protected and the research use and storage of their anonymized data on a public research data repository. Participants were told that by participating they would help scientists understand how and why different people think and feel about money in different ways. As an incentive for participation, they were offered (automated) video and web feedback on key self-reported financial capability measures, and their score on a financial knowledge test on completion of the online questionnaire, followed by a video of a television presenter offering them tips on personal financial management.

The first screen of the survey gave details of the study, anonymization of the data, the procedure for withdrawing consent and the use of the data in research. The survey required indication of informed consent before proceeding. Participants informed consent was indicated by ticking a box to agree to participate subject to the information they had read on the use of their data. Participants failing to agree were exited from the survey. Initial screening by the BBC excluded participants under the age of 16 from registering for surveys in this series. This is the age from which UK law assumes capacity to consent on a range of matters including to medical treatment and research such as clinical trials without parental involvement. All participants were provided with the email address of the BBC data controller and advised that they could withdraw their consent by contacting the data controller.”

There are no grants or grant numbers associated with this study. Just the support in kind from the BBC that we note.

4. In your Data Availability statement, you have not specified where the minimal data set underlying the results described in your manuscript can be found. PLOS defines a study's minimal data set as the underlying data used to reach the conclusions drawn in the manuscript and any additional data required to replicate the reported study findings in their entirety. All PLOS journals require that the minimal data set be made fully available. For more information about our data policy, please see http://journals.plos.org/plosone/s/data-availability. Upon re-submitting your revised manuscript, please upload your study’s minimal underlying data set as either Supporting Information files or to a stable, public repository and include the relevant URLs, DOIs, or accession numbers within your revised cover letter. For a list of acceptable repositories, please see http://journals.plos.org/plosone/s/data-availability#loc-recommended-repositories. Any potentially identifying patient information must be fully anonymized.

We have now clarified that the full data set, which includes the minimal data set is available on the UK data service repository, subject to free registration and some restrictions on use – the data set is well documented in line with the repository’s requirements. The restrictions on use are set by the data owner, the BBC, and are the same restrictions on use that were imposed by the BBC on the original study.

As we now clarify in the Ethics and data availability section of the paper:

“The full dataset is deposited with the UK data service, at https://doi.org/10.5255/UKDA-SN-8132-1. The owner of the IP in the dataset, the BBC, have imposed some conditions on data access which were also imposed on the original researchers. These are:

“The Data Collection is available to users registered with the UK Data Service, [registration is free]. Access is limited to applicants based in HE/FE institutions, for not-for-profit education and research purposes only. The BBC Data Collection shall not be used in a manner which:

• distorts the original meaning of the BBC Data Collection, for example by changing the context;

• discriminates against any specific social group or otherwise exploit vulnerable sections of society;

• promotes, encourages or facilitates violence;

• promotes, encourages or facilitates illegal activity;

• promotes, encourages or facilitates terrorism or other activities which risk national security;

• promotes the tobacco, armaments, alcohol or pornographic industries;

• encourages hatred on grounds of race, religion, gender, disability, age or sexual orientation;

• promotes, encourages or facilitates anti-social behaviour;

• might be perceived as damaging the BBC's reputation for accuracy or impartiality; or

• otherwise brings the BBC into disrepute.”

Users of the data set are subject to identity verification and are required to agree to the conditions of use.

The data set contains no identifying information on participants such as name, address, IP address, specific dates or contact information. The full data set does contain partial postcode data (last two characters are truncated), sufficient to identify postcode sector, which would typically cover around 3000 addresses.”

*We did not mean to imply that the data are available on request from the authors. We have now clarified the data access situation (see above).

*We have now clarified that the modest conditions on data reuse are imposed by the data owner, the BBC, and that these same conditions were imposed on the original research.

*Any minimal dataset we created would be subject to the same conditions imposed by the data owner, and its deposit would require a lengthy and onerous negotiation with them. We have taken the view that since the minimal data set is available as part of the well documented full data set on a public repository ( listed as acceptable to PLOS One)and subject only to modest conditions of use, that this should be sufficient.

REVIEWER COMMENTS 

Reviewer #1: 

The paper is generally well written, and will make an important contribution to the field. There are however some areas where a few changes/adjustments can further strengthen the contribution of this manuscript. 

*Thankyou.

This paper examined the association between money attitudes, financial capabilities, impulsiveness, and wealth accumulation using a large sample of U.K. based respondents. The results revealed that demographic differences accounted for 27%, money attitudes accounted for 14%, and financial capability accounted for 24% of the variances in wealth accumulation. The authors also found that planning ahead, age, and income were consistently significant predictors of wealth.

Although, the findings from this study are rather incremental than novel, the study confirms the findings from past literature (that were mostly conducted with small primary datasets), by empirically testing the hypotheses, using a large-scale dataset. I think the value of this research and this important contribution of the study has been undersold, and the authors need to highlight this significant contribution of their research.

*Thank-you for this suggestion. In the Discussion section we now note that “Although the findings are somewhat incremental in relation to prior studies, the size of the sample adds additional weight to previous findings. The ability to differentiate effects of financial capability, buying impulsiveness and money attitudes on different forms of wealth is also an important contribution to debates about the correlates of wealth accumulation.”

Comments and Suggestions

Analysis

Considering the heterogeneity of the readership of this journal, mentioning the purpose of reporting Hotelling’s Trace (Hotelling-Lawley) in the regression analysis will be informative to the readers.

*We have now added additional explanation: 

“The Hotelling’s trace value (also known as Hotelling-Lawler trace) is used to test the null hypothesis for each IV that it explains zero variance in the DVs. This statistic can be shown to generalize the partial F test used for ordinary least squares regression, with one dependent variable, to a test for regression with multiple dependent variables.”

Low Chronbach’s alpha for ‘Choosing products’ and ‘staying informed’. Some justification on why it maybe fine to use ‘choosing products’, ‘staying informed’, and the money attitude variables, inspite of their low alpha values will be useful.

We have now added a comment on this in the discussion of limitations at the end of the paper: -

“This, like all studies had various limitations we are conscious of. First, some financial capability and money attitude measures had modest Cronbach alpha reliability scores. Internal reliability tends to be lower for shorter scales since they tend to have greater random error of measurement. Larger more heterogeneous sample sizes also generate lower reliabilities (perhaps reflecting greater ease of tailoring item language to smaller more homogeneous samples [42]. 

The scales were measured as part of a large omnibus survey which had to trade off scale length for response rate and number of constructs measured. Clearly greater random measurement error (noise) can reduce statistical power of tests. However, given the large sample size, this is perhaps of less concern in the present study.”

Page 20

“We did not find support in the regression analysis for our third hypothesis, about buying impulsiveness, though all the correlations were negative as hypothesised. Impulsivity in

all spheres is associated with immaturity, poor emotional regulation and negative social outcomes [41]. It may be that effects of buying impulsivity on wealth are indirect for example accounted for via planning ahead and choosing products with which it correlates”.

But in Table 3, ‘buying impulsiveness’ has a positive beta 0.01 (p<0.01) for savings and other financial investments, and a beta of 0.02 (p<0.01) for value of physical items. Doesn’t this indicate that the negative association (as hypothesized) between impulsiveness and wealth accumulation goes away when other control variables are included in the model? Some clarification from the authors will be useful here.

We conducted a supplementary analysis as a check and now include the following in our results section:

“Given the discrepancy between correlational results and regression results, and noting the correlations of buying impulsivity with financial capabilities and attitudes, we conducted a supplementary multivariate multiple regression analysis entering buying impulsiveness with just the control variables. The results showed significant negative regression parameters for buying impulsiveness for property wealth (-.02**) and savings/investments (-.07***) but no significant parameter for physical items. This suggests buying impulsiveness may be associated with reduced investment in property and other savings, but with the effect masked by the variance it shares with financial capability and attitudes.”

Contribution of this study

Although the findings from this study are consistent based on the expectations and findings drawn from prior literature, and the contribution of this study appears incremental rather than novel; the results run with an incredibly rich dataset of over 90,000 respondents does provide validity to the findings from previous studies. I believe this is an important contribution of this study.

*As noted above we have use your suggestion to strengthen our claims for contribution.

Additional Limitations and Future Directions Potential Endogeneity

In spite of the rich data that has been used, it is still cross-sectional, and therefore the causality between variables, for example, ‘making ends meet’ and ‘saving and other financial investments’ may not be completely determined. This is because financial hardship or low savings might be driving the inability to make ends meet rather than not being able to make ends meet predicting lower levels of savings and other financial investments. This type of causality can be further analyzed using panel data analysis. I suggest that the authors include the use of cross- sectional dataset (instead of panel data) as a limitation of their study.

*We have added the following to the discussion of limitations:

“Third, the cross-sectional nature of the data, does not allow attributions of causality. For example, it is not possible to rule out reciprocal relationships between ability to make ends meet and wealth. On the one hand making ends meet should increase availability of funds to invest, conversely though having savings as a buffer may reduce the need to rely on credit. Future studies might usefully collect longitudinal panel data to explore the causal patterns in these variables.”

Possibility of Geospatial Analysis

The authors mention that geocodes were available for this dataset? If this is the case, then perhaps in future more research can be done by controlling for neighborhood/regional characteristics using the geospatial correlates. Although, this was not the purpose of this study, I think it will be beneficial to inform the readers of this possibility for those who maybe interested in pursuing this line of research in the future.

*We have now added the following in the discussion section: 

“For future users of this dataset, there is scope to extend the range of questions explored by exploiting the partial postcode geolocation data (at UK postal sector level) it includes this offer the possibility of combination with data, for example, on measures of relative social deprivation in different areas.” 

Potential for Generalizability

Although, the study uses a large dataset, it is a data of U.K. respondents. But, there is potential to replicate this study in other parts of the world, and this can be suggested in the future directions section of this manuscript.

*We have addressed these two points in the final section.

OTHER MINOR POINTS

Page 12: Results

“The correlations between all independent variables and the three dependent (wealth) variables are significant at the p< .5 level with most significant at p<.001”.

I believe this is a typographical error, and the authors really mean p<0.05 instead? 

*Yes, now corrected.

The paragraph indent on page 20, and a few other places do not seem correct. Overall, another round of copy editing will benefit this manuscript. 

*We have reformatted the paper and carried out further copy editing.

Reviewer #2: 

Overall, I felt that the authors have put together a pretty compelling manuscript that provides some good story lines. Here are some comments that might help push this manuscript along.

*Thank-you.

You might get some pushback on the notion that financial capability and financial literacy amount to the same thing, as you do on page 3 of your manuscript. Can a financially capable person not be very financially literate as experts measure the concept?...I would probably think so. Can a financially literate person be not very financially capable in a lot of settings, most likely due to low financial means?...I would think most definitely. You might want to check with some of the fairly nuanced literature on this subject matter through financial planning journals that are out there.

*We now say in the introduction: 

“There is also significant variation between authors in definitions of financial literacy [12]and financial capability, which are often taken to be synonyms or to significantly overlap [13]. For example, the OECD defines financial literacy as “A combination of financial awareness, knowledge, skills, attitudes and behaviours necessary to make sound financial decisions and ultimately achieve individual financial well-being” [14], whilst the Financial Capability Strategy for the UK, defines financial capability as “people’s ability to manage money well, both day to day and through significant life events, and their ability to handle periods of financial difficulty”. Drawing on a systematic review Goyal and Kumar (13: p81) suggest that “[p]eople can be financially literate when they have knowledge, understanding and skills to take care of their personal finances but they cannot be called financially capable unless it is reflected in their actual behaviour”; and “financial capability is the manifestation of a specified level of financial literacy and the execution of a desired financial behaviour” (p95).”

While I don’t have any quibbles with this author’s current lit review sources, there are a number of authors in the U.S. who specialize in the concept of financial literacy and personal finance that could probably serve as references here. Regarding the former, please check out some of the work from Brenda Cude and Gianni Nicolini (and many others). Some of the journals that you might want to scour for more perspective include the Journal of Consumer Affairs, Journal of Personal Finance, Journal of Financial Planning and Counseling, and Financial Planning Review. Two other book resources that should prove helpful for you are Degruyter’s Handbook of Personal Finance and the Routledge Handbook of Financial Literacy.

Thankyou for the suggestions, this was helpful in identifying a paper by Nicolini, Cude, and Chatterjee that was useful in our discussion of inconsistency of research definitions of financial literacy.

On page 5, you mention that “many popular writers have attempted to differentiate…” without providing a list of multiple references. Please fix this.

*Now fixed.

I would also make sure that all of your statements of fact are backed up by citations. I caught a few of them, but they are probably too numerous to enumerate in this space. For example, check out the top of page 7 with the statements “we know that…” These are practically begging for citations.

*Whilst being mindful of not over referencing, we have sought to identify passages that may give concern and added appropriate references.

While I am OK with the Hypothesis 2a on page 8, the write-up leading up to the hypothesis throws me off just a little bit. It seems that you are throwing a lot of words and concepts in there that are to be associated with “financial attitudes” that I might question.

*We have revised this passage for greater clarity and corrected the term ‘financial attitudes’ to ‘money attitudes’ to align with our earlier discussion.

Pretty impressive data set, and one that I admit I am not familiar with. Obviously the amount of data used is large enough to produce some interesting results. I am a little confused in the language on page 8 regarding the research methods, specifically regarding the data. While you refer to it as a secondary data set, you say that it required “informed consent”. This would be unlike a lot of large secondary data sets that exist in the U.S. Can you elaborate further? 

*The increasing prevalence of data repositories means that secondary analysis of a wide range of datasets is now becoming possible. This is not least because government research funding agencies in many countries (like the UK research councils) are both requiring deposit of data in repositories and encouraging their reuse. We are drawing on a data set that was deposited in this way with appropriate documentation of methods and ethical approaches, including informed consent. We report on the informed consent process from the original study.

BTW, kudos for mentioning that “under sampling amongst those with very low income and the elderly”. I would have guessed that, but you performed due diligence in pointing that out to your audience.

Thank you.

You mention “five bands” of income within the data set, but you list up to eight in your descriptive example. Please elaborate further or correct that item.

*Thank you for spotting this. It should say eight bands and we have corrected this.

The scale you mention on pages 10 to 11 appears to be more than 16 items, although I could be wrong about that. Please check though, just to make sure. As an aside, I think that a table detailing the development of the scales might be a worthwhile venture here.

We can confirm that the money attitudes scale has 16 items. This scale has been developed in prior research and used in multiple prior published studies. In such cases it is normal to reference the original study for full information on the scale and its development. 

Regarding some of your findings, there are a few things that do not seem to add up intuitively. The high correlation between making ends meet and savings/investment does not seem to add up. If you are scrounging to make a living month to month, how would you have enough to invest and put away?

*Making ends meet is scored such that higher scores mean doing better at making ends meet (note the example item is reverse scored). For example, higher scores mean less propensity to run out of money before the next pay day. Thus, it makes sense that higher scores on this variable are associated with greater capacity to invest money. Nonetheless we have added a sentence to clarify the direction of scoring.

 I do like your insights on the keeping track front, namely that wealthy folks may not need to track given their situation.

*Thank - you

I am not entirely familiar with the analysis you have conducted in Table 2. Can you possibly explain this methodology a little more? I have never worked with a model that I had to explain partial eta squared measures or Hotelling’s trace values.

*We have now added additional explanation: “The Hotelling’s trace value (also known as Hotelling-Lawler trace) is used to test the null hypothesis for each IV that it explains zero variance in the DVs. This statistic can be shown to generalize the partial F test used for ordinary least squares regression, with one dependent variable, to a test for regression with multiple dependent variables.” The partial eta squared scores are simply a convenient way to compare the unique variance explained by each variable. We note in the paper that “The partial eta squared values are effect sizes that provide a comparative indication of unique common variance between each IV and the DVs. Partial eta squared (η2p) values may be understood as the proportion of unique variance in the DVs explained by each IV once all other modelled relationships are controlled for [45]. We use partial eta squared measures in this study as an indicator of the relative importance of each IV in explaining the DVs.”

Does the inverse association between property wealth and education mean that educated people know not to invest in the property market? That finding would raise a few eyebrows, I would imagine.

*Since the correlation between these variables is modest but positive, it seems likely that the modest negative association in the regression is due to the main effect of education acting indirectly via another variable such as income with which it correlates (p=.30).

Be careful with using the following statement in your discussion, “Older, better educated males in higher income households had most wealth”. This would imply that you are using an interactive variable combining all three elements, when in truth you are only looking at them individually.

*Corrected.

I am not so sure that it is surprising that gender played as big a role as it did in wealth building. This could be confirmed by canvassing a lot more of the financial planning literature as I suggested earlier. There should be some good gender-related research on this kind of subject matter. There are still some pretty entrenched male-dominated financial planning gender roles within households that have endured throughout decades. Family and consumer sciences literature can attest to that as well.

*On reflection we agree and have removed this phrasing.

I would not beat yourself up over the lack of findings regarding the impulsiveness angle. It is an interesting facet of personal finance. Sometimes not finding statistically results is a story unto itself.

*See our earlier note on the supplementary analysis we carried out.

While you have referred to financial education, you have not really referred to financial planners or financial counselors, which are professions that have a toehold within the U.S. I would do some research on those professional lines, because this particular research has some implications for these professions.

*We have added these.

One other line of research that you might want to address is that of financial risk tolerance, which seems to have some relevance with some of your scalar sub-items. John Grable is a researcher that has written a ton on this front. I am specifically thinking along the impulsivity front, and its element of financially risky behavior.

*We think this is an important area and have done research in this area ourselves. However we do not directly measure risk-taking or risk tolerance and prefer to focus on the most direct implications of our findings.

There are some typos throughout the document. These don’t distract from the author’s generally good writing style, but it might not hurt to have a copy editor go through the document all the same.

*We have carried out further proof reading

---

## [Decision Letter · Decision Letter 1]

9 Nov 2022

Money attitudes, financial capabilities, and impulsiveness as predictors of wealth accumulation.

PONE-D-22-06524R1

Dear Dr. Fenton-O'Creevy,

We’re pleased to inform you that your manuscript has been judged scientifically suitable for publication and will be formally accepted for publication once it meets all outstanding technical requirements.

Kind regards,

Andrew T. Carswell

Academic Editor

PLOS ONE

Additional Editor Comments (optional):

Reviewers' comments:

Reviewer's Responses to Questions

**Comments to the Author**

1. If the authors have adequately addressed your comments raised in a previous round of review and you feel that this manuscript is now acceptable for publication, you may indicate that here to bypass the “Comments to the Author” section, enter your conflict of interest statement in the “Confidential to Editor” section, and submit your "Accept" recommendation.

Reviewer #1: All comments have been addressed

Reviewer #2: All comments have been addressed

2. Is the manuscript technically sound, and do the data support the conclusions?

Reviewer #1: Yes

Reviewer #2: Yes

3. Has the statistical analysis been performed appropriately and rigorously? 

Reviewer #1: Yes

Reviewer #2: Yes

4. Have the authors made all data underlying the findings in their manuscript fully available?

Reviewer #1: (No Response)

Reviewer #2: Yes

5. Is the manuscript presented in an intelligible fashion and written in standard English?

Reviewer #1: Yes

Reviewer #2: Yes

6. Review Comments to the Author

Reviewer #1: I am happy with the revised version of the manuscript. I believe the authors have addressed all of my comments and concerns regarding the paper.

Reviewer #2: Thank you for addressing my earlier comments and concerns. It appears that you have made sufficient improvements, for which I am most grateful.

7. PLOS authors have the option to publish the peer review history of their article (what does this mean?). If published, this will include your full peer review and any attached files.

Reviewer #1: No

Reviewer #2: No

---

## [Editor Report · Acceptance letter]

11 Nov 2022

PONE-D-22-06524R1 

Money attitudes, financial capabilities, and impulsiveness as predictors of wealth accumulation. 

Dear Dr. Fenton-O'Creevy:

I'm pleased to inform you that your manuscript has been deemed suitable for publication in PLOS ONE. Congratulations! Your manuscript is now with our production department. 

Kind regards, 

on behalf of

Dr. Andrew T. Carswell 

Academic Editor

PLOS ONE